## RESEARCH ARTICLE

# A PDZ-RapGEF promotes synaptic development in *Caenorhabditis elegans* through a Rap/Rac signaling pathway

Reagan Lamb*, Michael Scales, Julie Watkins, Martin Werner and Salvatore J. Cherra, III‡

## ABSTRACT

Small G proteins coordinate the development of nerve terminals. The activity of G proteins is finely tuned by GTPase regulatory proteins. Previously, we have observed that PXF-1, a *Caenorhabditis elegans* GTPase regulatory protein, is required for the function of cholinergic motor neurons. Here, we investigated how PXF-1 coordinates the development of presynaptic terminals at the molecular level. We observed that PXF-1 acts through RAP-1 to promote synapse development. Subsequently, we found that *pxf-1* mutants display a reduction in RAC-2 activity, which is required for cholinergic synapse development. We observed that RAC-2 acts downstream of RAP-1. Finally, we identified a physical interaction between RAP-1 and TIAM-1, a Rac guanine exchange factor, which links PXF-1 function to the presynaptic actin cytoskeleton through RAC-2 activation. These findings highlight how small G protein signaling pathways interact to coordinate the development of presynaptic terminals.

KEY WORDS: Neuromuscular junction, Synaptic vesicles, RapGEF, FRET-FLIM, G proteins

## INTRODUCTION

Synapses mediate the communication between presynaptic neurons and their postsynaptic targets, such as other neurons or muscle cells. During development, presynaptic terminals accumulate the necessary equipment for neurotransmission, such as active zone scaffolds, voltage-gated calcium channels, synaptic vesicles and exocytosis machinery. The formation of active zones and clustering of synaptic vesicle pools require the assembly of presynaptic actin filaments (Chia et al., 2014, 2012; Zhang and Benson, 2001). One set of proteins that modulate actin filaments is the family of GTPases known as small guanine nucleotide-binding proteins (small G proteins). The family of small GTPases is comprised of five subfamilies: Rho, Ras, Arf, Rab and Ran. Some subfamily members, like Rho, Rac and Rap, modulate the actin cytoskeleton. In mammalian neurons, Rac proteins influence dendrite development, synapse function and memory (Cheng et al., 2021; Rao et al., 2019). In the *Caenorhabditis elegans* nervous system, Rac homologs regulate axon guidance and promote synaptic vesicle clustering (Lundquist

Department of Neuroscience, University of Kentucky College of Medicine, 780 Rose St, Lexington, KY 40536, USA.
*Present address: Department of Genetics, Yale University School of Medicine, Sterling Hall of Medicine, 333 Cedar St, New Haven, CT 06510, USA.

‡Author for correspondence (scherra@uky.edu)

R.L., 0000-0002-3390-401X; S.J.C., 0000-0003-1581-9150

et al., 2001; Stavoe and Colón-Ramos, 2012). Other members, like Rap proteins, ensure proper neuronal migration, synaptic transmission and plasticity (Franco et al., 2011; Jossin and Cooper, 2011; Morozov et al., 2003; Pan et al., 2008; Subramanian et al., 2013). In *C. elegans*, the Rap homolog RAP-2 restricts synapse formation to ensure proper tiling of presynaptic sites in motor neurons (Chen et al., 2018). Overall, small G proteins play essential roles during nervous system development.

The activity of small G proteins is coordinated by GTPase activating proteins (GAPs) and guanine nucleotide exchange factors (GEFs). In general, GAPs reduce G protein signaling by stimulating the hydrolysis of GTP by the G protein, which leads to its inactivation. GEFs accelerate the exchange of GDP for GTP, which stimulates G protein signaling. RasGAPs can act on both Ras and Rap to limit their signaling activity and are essential for synapse formation, dendrite development and synaptic plasticity (Araki et al., 2020; Chen et al., 2018; Duan et al., 2014; Vazquez et al., 2004). PDZ-containing RapGEFs promote Rap signaling to ensure proper neuronal development and synaptic function. For example, RapGEF2 or RapGEF6 knockout mice display deficits in neural progenitor development (Maeta et al., 2016). Separately, RapGEF6 was shown to modulate neuronal activity and synaptic plasticity in various brain regions (Levy et al., 2015). In *C. elegans*, we have found that the PDZ-GEF, PXF-1, is required for synaptic development through an actin-mediated mechanism (Lamb et al., 2022).

In this study, we further investigated how PXF-1 functions to modulate the abundance of synaptic vesicles at presynaptic terminals using the *C. elegans* neuromuscular junction (NMJ). We found that mutations in the genes encoding two small G proteins, *rap-1* or *rac-2*, reduced synapse development in cholinergic motor neurons, as was seen in *pxf-1* mutants. Activating mutations in either RAP-1 or RAC-2 were sufficient to restore synapse development in *pxf-1* mutants. Additional data indicate that RAP-1 acts upstream of RAC-2 and its GEF, TIAM-1, to promote synapse development. This study has uncovered a PXF-1 signaling pathway that sequentially activates two small G proteins to maintain presynaptic actin filaments and promote the accumulation of synaptic vesicles at cholinergic terminals.

## RESULTS

### PXF-1 functions in cholinergic neurons to promote the accumulation of synaptic vesicles during development

We have previously observed that PXF-1 is required for normal levels of synaptic vesicle markers in cholinergic neurons at the NMJ (Lamb et al., 2022). As *pxf-1* mutants produce synapses that contain a dimmer fluorescent signal from synaptic vesicle markers, we sought to determine if this reduction in synaptic vesicle markers was associated with a delay in synapse development. Using mCherry::RAB-3 expressed in cholinergic neurons under the *unc-129* promoter (Zhou et al. 2017), we observed increased fluorescence during development in wild-type and *pxf-1* animals (Fig. 1A,B). However, the mCherry::RAB-3 signal was much lower

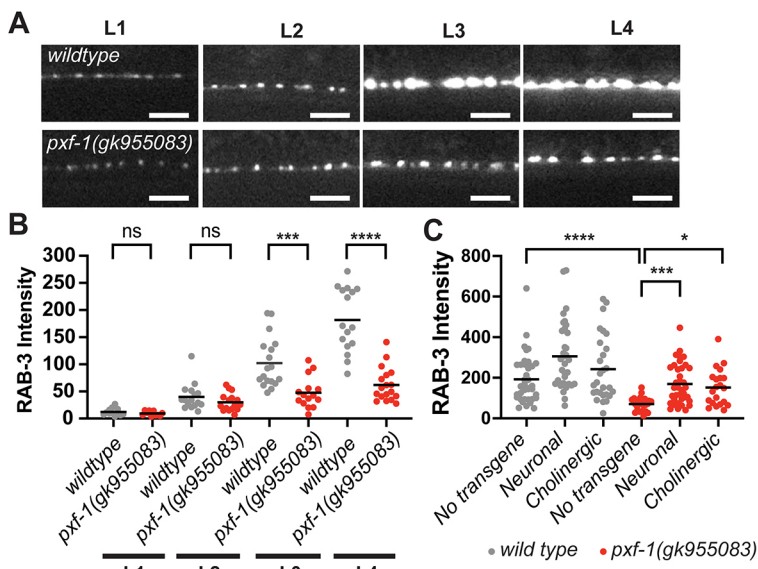

**Fig. 1. Mutant *pxf-1* animals display a deficiency in the accumulation of RAB-3-labeled vesicles during development.** (A) Representative images of dorsal cord synapses labeled by mCherry::RAB-3 expressed under the *unc-129* promoter in wild-type and *pxf-1(gk955083)* mutant animals during larval stages: L1, L2, L3 and L4. Scale bars: 5 µm. (B) Quantification of mCherry::RAB-3 grayscale fluorescence intensity. Gray circles represent individual wild-type animals, red circles represent individual *pxf-1(gk955083)* mutants. Black bars indicate means. *n*=12-18. ****P*<0.001, *****P*<0.0001 (Kruskal–Wallis with Dunn's multiple comparison correction). (C) Quantification of mCherry::RAB-3 grayscale fluorescence intensity in wild type and *pxf-1(gk955083)* mutants expressing *pxf-1* cDNA in all neurons, only cholinergic neurons or no transgene. For pan-neuronal expression, *bluEx53* and *bluEx54* analyses were combined into a single dataset named Neuronal, and for cholinergic expression, *bluEx59* and *bluEx61* analyses were combined into a single dataset named Cholinergic. Gray circles represent individual wild-type animals, red circles represent *pxf-1(gk955083)* mutants. Black bars indicate means. *n*=21-37. **P*<0.05, ****P*<0.001, *****P*<0.0001 (Kruskal–Wallis with Dunn's multiple comparison correction). ns, not significant.

in *pxf-1* mutants compared with wild-type animals (Fig. 1A,B). We observed a lower fluorescence signal as early as L2 animals, and this reached statistical significance by L3 and L4 stages.

We previously observed that PXF-1 functions in neurons but not muscle to promote neuromuscular function (Lamb et al., 2022). To determine in which neurons PXF-1 functions to promote the accumulation of mCherry::RAB-3-labeled synaptic vesicles, we first expressed wild-type *pxf-1* cDNA in all neurons using the *rgef-1* promoter. We found that pan-neuronal expression of *pxf-1* cDNA was sufficient to increase mCherry::RAB-3 in *pxf-1* mutant animals (Fig. 1C). To determine if *pxf-1* expression solely in cholinergic neurons is sufficient to restore proper synapse development, we used the *unc-17b* promoter to drive *pxf-1* expression. We observed that expression of *pxf-1* in cholinergic neurons also was sufficient to increase cholinergic mCherry::RAB-3 levels in *pxf-1* mutant animals (Fig. 1C). Together, these data suggest that PXF-1 functions in a cell autonomous manner to promote the development of cholinergic NMJs.

### RAP-1 activation restores synaptic vesicle intensity in *pxf-1* mutants

To determine which G proteins mediate the effects of PXF-1 on synapse development, we next investigated the Rap homologs *rap-1* and *rap-2*, which function with PXF-1 to promote epithelial development (Pellis-van Berkel et al., 2005). We measured the intensity and density of cholinergic NMJs labeled by mCherry::RAB-3 in previously identified *rap-1* and *rap-2* mutant animals (Pellis-van Berkel et al., 2005; Thompson et al. 2013). We found that only *rap-1* mutants displayed a decrease in mCherry::RAB-3 intensity, but *rap-2* mutants were indistinguishable from wild-type animals (Fig. 2A-D). We also observed a small but significant increase in the number of RAB-3-labeled puncta in *rap-1* mutant animals (Fig. 2E).

If RAP-1 is the target of PXF-1, we would expect that RAP-1 signaling would be reduced in *pxf-1* mutants. Therefore, we hypothesized that activating RAP-1 signaling in cholinergic neurons should restore mCherry::RAB-3 intensity in *pxf-1* mutant animals. To increase RAP-1 signaling in cholinergic neurons, we expressed a constitutively active RAP-1(G12V) cDNA under the *unc-17b* promoter in the *pxf-1(gk955083)* mutants expressing mCherry::RAB-3. This constitutively active mutation blocks hydrolysis of the bound GTP and prevents the G protein from returning to an inactive state (Hammond et al., 2015). To control for increased levels of

expression of RAP-1 in the transgenic animals, we also generated a cholinergic RAP-1(WT) cDNA transgene. We found that expression of the constitutively active RAP-1(G12V) cDNA was able to restore the intensity of mCherry::RAB-3 in *pxf-1* mutant animals to wild-type levels (Fig. 2F). However, expression of the RAP-1(WT) cDNA transgene did not increase the intensity of mCherry::RAB-3-labeled puncta in *pxf-1* mutant animals (Fig. 2F). These data indicate that exogenous activation of RAP-1 signaling can bypass the requirement for PXF-1 for synapse development and suggest that RAP-1 functions downstream of PXF-1.

### Mutations in *rac-2* reduce aldicarb sensitivity, synaptic vesicle intensity and presynaptic F-actin levels

In a previous study investigating how PXF-1 promotes synapse assembly, we found that neuronal expression of WVE-1 was sufficient to restore synapse development in *pxf-1* mutants (Lamb et al., 2022). WVE-1 is the *C. elegans* homolog of the evolutionarily conserved WAVE (WASP-family verprolin homolog) or SCAR (suppressor of cyclic AMP receptor) complex. Mutations in *wve-1* and other WAVE complex protein genes reduce presynaptic assembly in *C. elegans* and mammals (Chia et al., 2014; Hazai et al., 2013; Stavoe and Colón-Ramos, 2012; Stavoe et al., 2012). The WAVE complex is activated by Rac GTPases (Miki et al., 1998; Westphal et al., 2000), but there is no evidence of Rap GTPases directly activating the WAVE complex (Fig. 3A). Therefore, we sought to determine whether a Rac homolog may be involved in PXF-1 signaling along with RAP-1. We used aldicarb sensitivity as an initial screen to identify Rac GTPases that may phenocopy the reduced sensitivity to aldicarb that was observed in *pxf-1* mutants (Lamb et al., 2022). Aldicarb is an acetylcholinesterase inhibitor that causes the accumulation of acetylcholine at the NMJ and results in paralysis. We previously observed that mutations in *pxf-1* reduce cholinergic activity that results in a slower rate of paralysis when incubated on aldicarb plates (Lamb et al., 2022). We found that mutations in two Rac GTPases, *ced-10(n1993)* and *rac-2(ok326)*, caused a significant decrease in aldicarb sensitivity, which is consistent with *ced-10* and *rac-2* mutations reducing cholinergic neurotransmission (Fig. 3B). Next, we examined synaptic vesicle abundance in *ced-10* and *rac-2* mutant animals by using immunofluorescence to measure the levels of endogenous UNC-17, the vesicular acetylcholine transporter (VAChT) in *C. elegans*. Previously, we have shown that mutations in

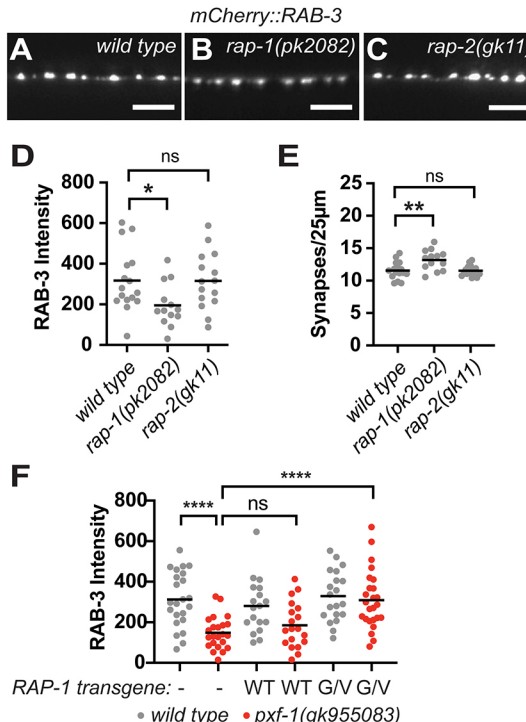

**Fig. 2. RAP-1 activation restores synaptic vesicle intensity in *pxf-1* mutants.** (A-C) Representative images of mCherry::RAB-3-labeled cholinergic synapses in the dorsal cord of (A) wild type, (B) *rap-1(pk2082)* and (C) *rap-2(gk11)*. Scale bars: 5 µm. (D) Quantification of mCherry::RAB-3 grayscale fluorescence intensity. (E) Quantification of synaptic density. Gray circles represent individual animals, black bars indicate means. *n*=13-16. **P*<0.05, ***P*<0.01 (one-way ANOVA with Dunnett's multiple comparisons correction in D; one-way ANOVA with Sidak's multiple comparisons correction in E). (F) Quantification of mCherry::RAB-3 grayscale fluorescence intensity from wild-type or *pxf-1(gk955083)* mutant animals expressing no transgene (labeled as a dash), RAP-1(WT)::mGFP (labeled as WT) or RAP-1(G12V)::mGFP (labeled as G/V) transgenes in cholinergic motor neurons. For RAP-1(WT), *bluEx143* and *bluEx144* were combined into one dataset, and for RAP-1(G12V), *bluEx146* and *bluEx147* were combined into one dataset. Each data point represents an individual animal. Gray circles are wild-type animals and red circles are *pxf-1(gk955083)* mutants. Black bars indicate means. *n*=18-25. *****P*<0.0001 (one-way ANOVA with Sidak's multiple comparisons correction). ns, not significant.

*pxf-1* reduced the intensity of UNC-17 stained puncta (Lamb et al., 2022). Here, we found that previously identified *ced-10(n1993)*, *rac-2(ok326)* and *rac-2(gk281)* mutant animals (Lundquist et al., 2001; *C. elegans* Deletion Mutant Consortium, 2012) also exhibited decreases in UNC-17 puncta intensity, suggesting a decrease in synaptic vesicle abundance (Fig. 3C-G).

While both CED-10 and RAC-2 GTPases displayed deficits in synaptic development and function, we chose to focus on RAC-2 for the remainder of this study, as the function of RAC-2 has been studied to a lesser degree than CED-10. To determine whether *rac-2* modulates the actin cytoskeleton, like *pxf-1* (Lamb et al., 2022), we quantified presynaptic filamentous actin (F-actin) in cholinergic neurons using a GFP tagged utrophin (ut). Ut is an F-actin binding protein, and its interaction with F-actin is mediated by its calponin homology (CH) domain (Rybakova and Ervasti, 2005; Winder et al., 1995). Previous studies have used GFP::ut-CH to evaluate F-actin levels (Burkel et al., 2007; Ladt et al., 2016; Patel et al., 2017), including studies focused on F-actin at presynaptic terminals in *C. elegans* (Chia et al., 2014; Stavoe and Colón-Ramos, 2012). We previously created a similar mGFP::ut-CH driven by the *unc-17b* promoter to label F-actin in cholinergic

neurons, which also expressed mCherry::RAB-3 under the *unc-129* promoter to visualize cholinergic synapses (Lamb et al., 2022). Using mGFP::ut-CH, we found that disruption of *pxf-1* reduces F-actin in cholinergic terminals (Lamb et al., 2022). Here, we observed a decrease in mGFP::ut-CH signal in mCherry::RAB-3-labeled synaptic terminals in *rac-2* mutant animals (Fig. 3H-N), suggesting that RAC-2 promotes the assembly or maintenance of presynaptic actin filaments. Overall, these findings are consistent with RAC-2 acting with PXF-1 to promote synapse development in cholinergic neurons in an actin-based manner.

### RAC-2 and PXF-1 act in the same pathway

While it is evident that *rac-2* mutant animals phenocopy *pxf-1* mutants, it is still unclear whether they are working together to coordinate synapse development. To investigate whether RAC-2 acted in the same pathway as PXF-1, we used animals expressing mCherry::RAB-3 in cholinergic neurons. We found that animals containing mutations in *pxf-1* or *rac-2* displayed a decrease in synaptic vesicle intensity compared to wild-type animals, but no change in synapse density (Fig. 4A-F). Additionally, *pxf-1(gk955083); rac-2(ok326)* double mutants showed a decrease in mCherry::RAB-3 intensity that was not significantly different from either single mutant (Fig. 4A-E), consistent with PXF-1 and RAC-2 acting in the same pathway.

### PXF-1 modulates RAC-2 activity

To determine if RAC-2 activity levels in cholinergic motor neurons were influenced by PXF-1 function, we expressed a fluorescence resonance energy transfer (FRET)-fluorescence lifetime imaging microscopy (FLIM) biosensor under the *unc-17b* promoter. Based on previous designs (Aoki and Matsuda, 2009; Chen et al., 2018; Graham et al., 2001), our biosensor expresses mCherry fused to the Rac binding domain (RBD) from *C. elegans pak-1*, a T2A ribosomal skip sequence, and mGFP fused to *rac-2* cDNA (Fig. 5A,B). When RAC-2 is activated, mGFP::RAC-2 binds to mCherry::RBD, which produces FRET between the mGFP donor and the mCherry acceptor and reduces the time mGFP exists in an excited state (Fig. 5A). Reductions in RAC-2 activity would reduce the level of FRET and increase the lifetime of mGFP (Fig. 5B).

We first tested whether our sensor was able to detect changes in RAC-2 activity. In neurons, Rac proteins are activated through their canonical Rac GEF, TIAM-1 (Brar et al., 2022; Demarco et al., 2012); therefore, we compared the lifetime of mGFP::RAC-2 between wild-type and *tiam-1(tm1556)* mutant animals. We found that the mean lifetime fluorescence of mGFP::RAC-2 in *tiam-1(tm1556)* animals was higher than in wild-type animals, indicating a decrease in RAC-2 activity, which is consistent with TIAM-1 function (Fig. 5C,D,F). Additionally, these changes in lifetime fluorescence are within the range of changes caused by biologically relevant events (Stubbs et al., 2005).

To determine if disrupting PXF-1 altered the level of RAC-2 activity, we compared mGFP::RAC-2 lifetime fluorescence in *pxf-1* mutants to wild-type animals. Similar to *tiam-1* mutant animals, we observed a higher mean lifetime for mGFP::RAC-2 in *pxf-1* mutants compared with wild-type animals (Fig. 5C,E,F). These data are consistent with PXF-1 promoting the activation of RAC-2 signaling in cholinergic neurons, which is decreased in *pxf-1* mutant animals, and suggest that RAC-2 may function downstream of PXF-1.

### RAC-2 acts downstream of PXF-1

To determine if RAC-2 acts downstream of PXF-1, we examined whether activation of RAC-2 was sufficient to rescue synaptic

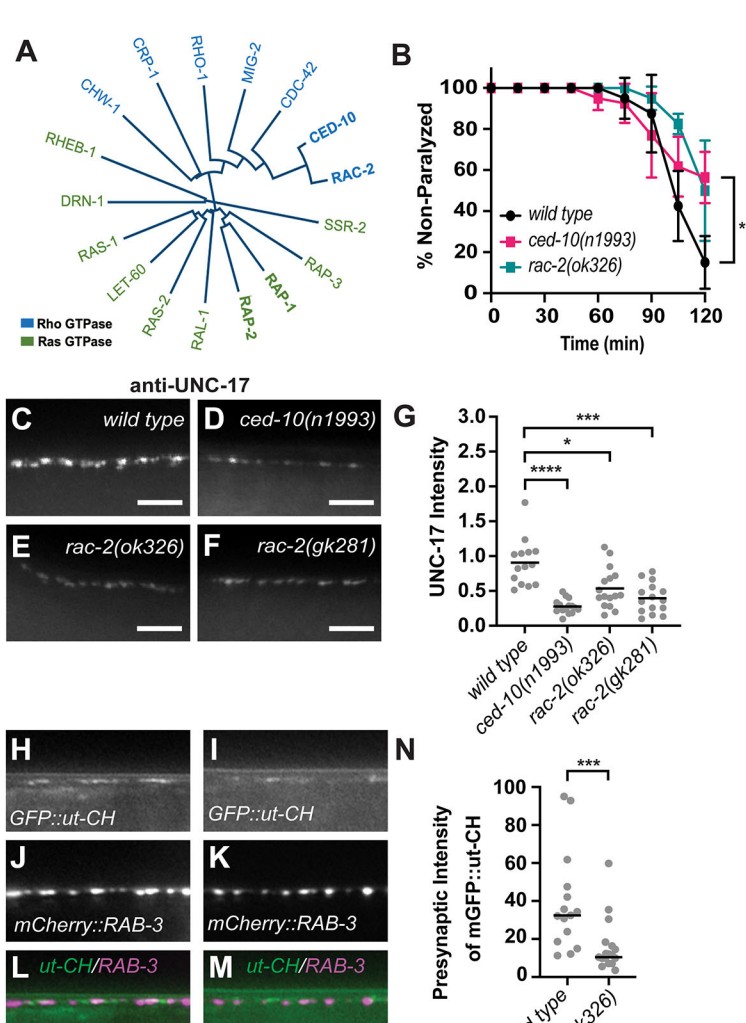

**Fig. 3. Mutations in *rac-2* reduce aldicarb sensitivity, synaptic vesicle intensity and presynaptic F-actin levels.**
(A) Phylogenetic tree of Rho and Ras G protein family members. (B) Quantification of non-paralyzed animals during exposure to 1 mM aldicarb. Symbols represent means; error bars represent s.e.m. *n*=5 trials of ten animals each. *P<0.05 (two-way ANOVA with Dunnett's multiple comparison correction). (C-F) Representative images of dorsal cord stained by UNC-17 antibodies in (C) wild type, (D) *ced-10(n1993)*, (E) *rac-2(ok326)* and (F) *rac-2(gk281)*. (G) Quantification of fluorescence intensity of UNC-17 staining in dorsal nerve cord. Gray circles represent individual animals, black bars indicate means. *n*=14-16. *P<0.05, ***P<0.001, ****P<0.0001 (Kruskal–Wallis with Dunn's multiple comparison correction). (H-M) Representative images of dorsal cord in wild-type and *rac-2(ok326)* mutant animals with (H,I) mGFP-labeled calponin homology domain of F-actin binding protein utrophin (GFP::ut-CH), (J,K) mCherry-labeled RAB-3 in cholinergic synapses and (L,M) merged channels. (N) Quantification of grayscale intensity of mGFP::ut-CH within 1 µm of the peak of mCherry::RAB-3 fluorescence signal. Gray circles represent individual animals, black bars indicate median. *n*=15-17. ***P<0.001 (Mann–Whitney test for non-parametric data). Scale bars: 5 µm.

vesicle accumulation defects. We expressed RAC-2(WT) or the constitutively activated RAC-2(G12V) cDNA in cholinergic neurons under the *unc-17b* promoter and measured the intensity of mCherry::RAB-3 in the wild-type and *pxf-1(gk955083)* mutant animals. Like RAP-1(G12V) mutant transgenes, expression of RAC-2(G12V) was sufficient to increase synaptic vesicle intensity in *pxf-1* mutants to wild-type levels (Fig. 6A). Unexpectedly, exogenous expression of RAC-2(WT) also increased mCherry::RAB-3 intensity in *pxf-1* mutants. Together, these data suggest that RAC-2 signaling functions downstream of PXF-1 to promote synapse development; however, our data with the RAC-2(WT) transgene suggest that compensatory factors may be primed to increase RAC-2 activity in *pxf-1* mutants.

### RAP-1 functions upstream of RAC-2 to promote synapse development

Activation of RAP-1 or RAC-2 was sufficient to restore synapse development in *pxf-1* mutant animals (Figs 2F and 6A). While these data indicate that RAP-1 and RAC-2 function in the same pathway as PXF-1, they do not provide insight into the hierarchy of these two G proteins in the signaling pathway. To determine the order of how RAP-1 and RAC-2 function to modulate synaptic development, we measured mCherry::RAB-3 levels in *rac-2* mutants expressing the RAP-1(WT) or RAP-1(G12V) transgene in cholinergic motor neurons. Expression of RAP-1(WT) or RAP-1(G12V) was not able

to restore mCherry::RAB-3 levels to wild-type levels in *rac-2* mutants (Fig. 6B). Since the same RAP-1(G12V) transgenes were sufficient to restore synapse development in *pxf-1* mutants (Fig. 2F) but did not rescue synapse development in *rac-2* mutants, these data are consistent with RAP-1 functioning downstream of PXF-1 but upstream of RAC-2.

### TIAM-1 functions with PXF-1 to restore synapse development

We next sought to determine how PXF-1 and RAP-1 signaling influences RAC-2 activity. Tiam1 and Vav3 have been identified as Rac GEFs that modulate synapse development or function (Duman et al., 2013; Ulc et al., 2017; Um et al., 2014). The *C. elegans* genome has homologs for both TIAM1, called *tiam-1*, and VAV3, called *vav-1* (Demarco et al., 2012; Duman et al., 2013). As *vav-1* mutants cause embryonic lethality, hypersensitivity to aldicarb, and function in an interneuron to control locomotor behavior (Etheridge et al., 2015; Fry et al., 2014), we decided to focus on *tiam-1*, which is required for RAC-2 activation (Fig. 5E). We measured mCherry::RAB-3 intensity in cholinergic neurons in two *tiam-1* single mutants (*tm1556* or *ok772*) and in double mutants with *pxf-1*. We found that both *tiam-1* alleles decreased mCherry::RAB-3 intensity (Fig. 7A-E,G-K). Double mutants between *tiam-1* and *pxf-1* displayed a reduction in mCherry:: RAB-3 intensity, but there were no significant differences between

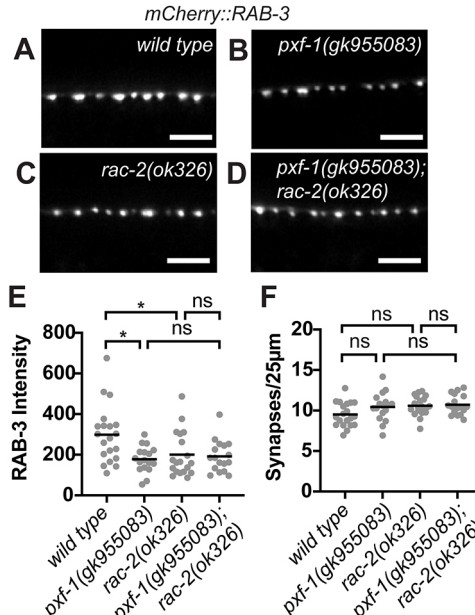

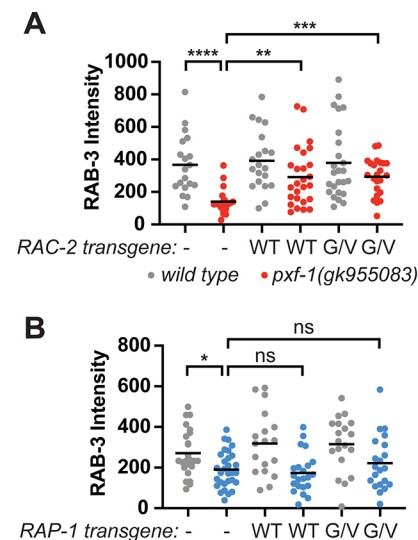

**Fig. 4. RAC-2 and PXF-1 act in the same pathway.** (A-D) Representative images of mCherry::RAB-3-labeled cholinergic synapses in the dorsal cord of (A) wild-type, (B) *pxf-1(gk955083)*, (C) *rac-2(ok326)* and (D) *pxf-1(gk955083); rac-2(ok326)* animals. Scale bars: 5 µm. (E) Quantification of mCherry::RAB-3 grayscale fluorescence intensity. Gray circles represent individual animals, black bars indicate means. *n*=17-20. *$P<0.05$ (Kruskal–Wallis with Dunn's multiple comparison correction). (F) Quantification of synapse density. Gray circles represent individual animals, black bars indicate means. *n*=17-20. One-way ANOVA with Sidak's multiple comparisons correction. ns, not significant.

the single and double mutants (Fig. 7A-E,G-K). There were no changes in synapse number in any single or double mutants (Fig. 7F,L). Together these data suggest that TIAM-1 and PXF-1 act in the same pathway.

### RAP-1 interacts with TIAM-1 in cholinergic motor neurons

Based on our data, we hypothesized that PXF-1 activates RAP-1 signaling and that TIAM-1 activates RAC-2 signaling downstream of RAP-1. Tiam1 in mammals contains a Ras-binding domain, which mediates interactions between small GTPases and their downstream effectors (Lambert et al., 2002). Although *C. elegans* TIAM-1 lacks a predicted Ras-binding domain, sequence alignment

**Fig. 6. RAC-2 functions downstream of PXF-1 and RAP-1 to promote synapse development.** (A) Quantification of mCherry::RAB-3 grayscale fluorescence intensity from wild-type or *pxf-1(gk955083)* mutant animals expressing no transgene (labeled as a dash), RAC-2(WT)::mGFP (labeled as WT) or RAC-2(G12V)::mGFP (labeled as G/V) transgenes under the *unc-17b* promoter. For RAC-2(WT), *bluEx166* and *bluEx168* were combined into one dataset and for RAC-2(G12V), *bluEx169* and *bluEx171* were combined into one dataset. Each data point represents an individual animal. Gray circles are wild-type and red circles are *pxf-1(gk955083)* mutant animals. Black bars indicate means. *n*=19-26. **$P<0.01$, ***$P<0.001$, ****$P<0.0001$ (Kruskal–Wallis with Dunn's multiple comparison correction). (B) Quantification of mCherry::RAB-3 grayscale fluorescence intensity from wild-type or *rac-2(ok326)* mutant animals expressing no transgene (labeled as a dash), RAP-1(WT)::mGFP (labeled as WT) or RAP-1(G12V)::mGFP (labeled as G/V) transgenes under the *unc-17b* promoter. For RAP-1(WT), *bluEx143* was used, and for RAP-1(G12V), *bluEx146* was used. Each data point represents an individual animal. Gray circles are wild-type and blue circles are *rac-2(ok326)* mutant animals. Black bars indicate means. *n*=18-30. *$P<0.05$ (one-way ANOVA with Sidak's multiple comparisons correction). ns, not significant.

among human, mouse and *C. elegans* Tiam1/TIAM-1 proteins displays ∼20% identity between *C. elegans* TIAM-1 and the Ras-binding domains from its mammalian homologs (Fig. 8A). Therefore, we hypothesized that TIAM-1 may be a direct effector of RAP-1 in cholinergic neurons. To test this hypothesis, we used

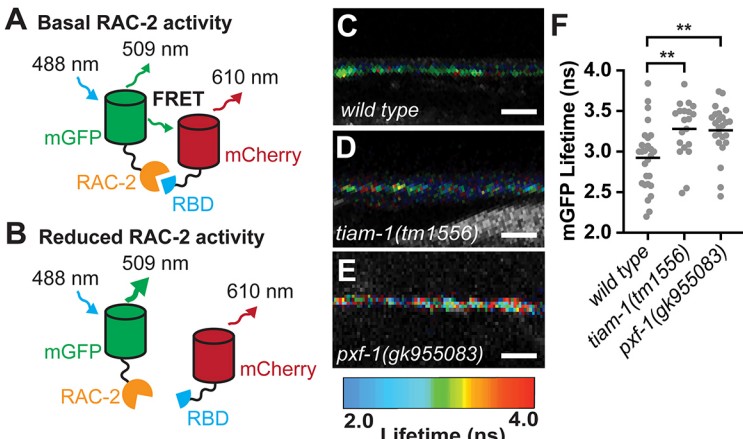

**Fig. 5. PXF-1 modulates RAC-2 activity.** (A,B) Schematic of biosensor function. In wild-type animals (A), basal levels of RAC-2 will lead to an interaction between RAC-2 and the Rac binding domain (RBD) from *pak-1*. This interaction will promote FRET between RAC-2::mGFP and mCherry::RBD proteins. If a mutant background reduces RAC-2 activity (B), there should be fewer interactions between RAC-2::mGFP and mCherry::RBD proteins, thus reducing FRET and increasing the lifetime of mGFP fluorescence. (C-E) FastFLIM images of mGFP in the dorsal cord of (C) wild-type, (D) *tiam-1(tm1556)* and (E) *pxf-1(gk955083)* animals expressing the biosensor in cholinergic neurons. mGFP lifetime has been pseudo-colored to represent fluorescence lifetime values in nanoseconds. Scale bars: 5 µm. (F) Mean lifetime fluorescence of RAC-2::mGFP in wild-type, *tiam-1(tm1556)* and *pxf-1(gk955083)* mutant backgrounds. Independent alleles for the RAC-2 biosensor (*bluEx71* and *bluEx72*) were used for each genotype; graph shows data that were pooled from *bluEx71* and *bluEx72*. Gray circles represent individual animals, black bars indicate means. *n*=18-25. **$P<0.01$ (one-way ANOVA with Dunnett's multiple comparisons correction).

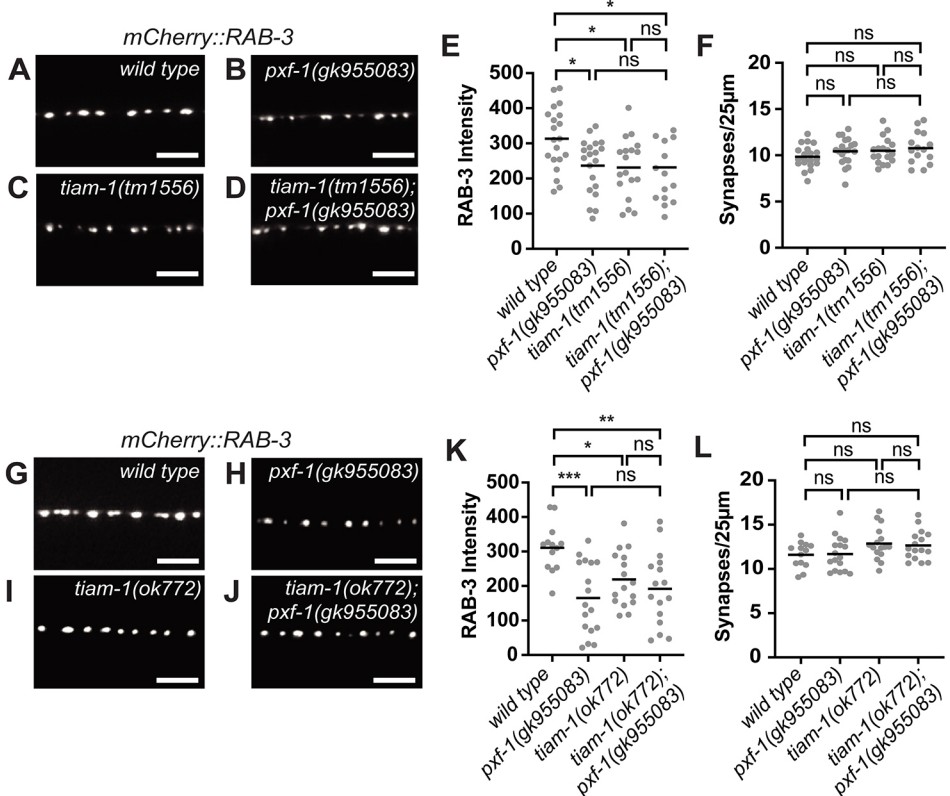

**Fig. 7. TIAM-1 functions with PXF-1 to restore synapse development.** (A-D) Representative images of mCherry::RAB-3-labeled cholinergic synapses in the dorsal cord of (A) wild-type, (B) *pxf-1(gk955083)*, (C) *tiam-1(tm1556)* and (D) *tiam-1(tm1556); pxf-1(gk955083)* animals. (E) Quantification of mCherry:: RAB-3 grayscale fluorescence intensity. (F) Quantification of synaptic density. Gray circles represent individual animals, black bars indicate means. *n*=15-20. *P<0.05 (one-way ANOVA with Sidak's multiple comparisons correction in E and F). (G-J) Representative images of mCherry:: RAB-3-labeled cholinergic synapses in the dorsal cord of (G) wild type, (H) *pxf-1(gk955083)*, (I) *tiam-1(ok772)* and (J) *tiam-1(ok772); pxf-1(gk955083)*. (K) Quantification of mCherry::RAB-3 grayscale fluorescence intensity. (L) Quantification of synaptic density. Gray circles represent individual animals, black bars indicate means. *n*=13-19. *P<0.05, **P<0.01, ***P<0.001 (one-way ANOVA with Sidak's multiple comparisons correction in K and L). ns, not significant. Scale bars: 5 μm.

FRET-FLIM to determine whether TIAM-1 and RAP-1 interact in cholinergic motor neurons. We expressed TIAM-1::mGFP with either mCherry or RAP-1::mCherry in cholinergic neurons and measured the lifetime fluorescence of mGFP (Fig. 8B,C). We observed a significant decrease in TIAM-1::mGFP lifetime in the presence of RAP-1:: mCherry as compared to free mCherry (Fig. 8D-F). This decrease in TIAM-1::mGFP lifetime fluorescence is consistent with an interaction between RAP-1 and TIAM-1.

## DISCUSSION

Synapse development relies on precisely coordinated signaling events to establish and maintain functional connections. Small G proteins can localize to specific regions of cells and oscillate between active and inactive states. Therefore, modulation of their localization or duration of activity represents one way to coordinate the growth and development of tissues. In the nervous system, Rap proteins influence synaptic connectivity and function by modulating the density of dendritic spines, abundance of neurotransmitter receptors, or numbers of synaptic terminals (Beaudoin et al., 2012; Chen et al., 2018; Heo et al., 2017; Ryu et al., 2008; Zhu et al., 2002). Here, we show that *C. elegans* RAP-1, a target of PXF-1, promotes the development of presynaptic terminals at the NMJ. Our data indicate that PXF-1 induction of RAP-1 signaling leads to the activation of RAC-2, which modulates the presynaptic actin cytoskeleton (Fig. 9). We propose that activated RAP-1 interacts with TIAM-1 to connect RAP-1 signaling to RAC-2 (Fig. 9); however, we have not mapped the precise protein domains that mediate the interaction between RAP-1 and TIAM-1.

In mammalian cells, a similar interaction between Rap and Rac GTPases promotes cytoskeletal regulation (Taira et al., 2004). Multiple studies outside the nervous system support a direct interaction between Rap1 and two Rac GEFs, Tiam1 or Vav2, to modulate actin filaments in a Rac-dependent manner (Arthur et al.,

2004; Birukova et al., 2013, 2008; Gérard et al., 2007). Similar pathways also underlie cortical development. For example, Rap1 acts upstream or parallel to Rac1 to promote neuronal migration (Jossin and Cooper, 2011). This study suggested that Vav2 activation is the most likely Rac GEF to connect Rap1 to Rac GTPase signaling during cortical neuron migration. Our study suggests that RAP-1 acts through TIAM-1 and RAC-2 to promote the accumulation of synaptic vesicles at presynaptic terminals (Fig. 9). However, we have not investigated which pathways may be involved in the increased synaptogenesis observed in *rap-1* mutants (Fig. 2E). As *tiam-1* and *rac-2* mutants do not display increases in the number of cholinergic synapses (Figs 4F and 7F,L), we would hypothesize that the effects of RAP-1 on synapse number may proceed through a different mechanism.

### Divergent roles of Rap GTPases at the synapse

Throughout evolution, most species contain at least two Rap paralogs, Rap1 and Rap2. While both paralogs facilitate similar biological processes, they appear to operate through different signaling mechanisms, or may even oppose one another. For example, Rap1 in mammals promotes dendritic spine elongation in cortical neurons, but Rap2 activation causes the formation of shorter spines in the forebrain of mice (Ryu et al., 2008; Xie et al., 2005). In hippocampal neurons, activation of Rap2, but not Rap1, reduces the complexity of axons and dendritic arbors; however, inhibition of Rap1 signaling resulted in shorter dendrites (Fu et al., 2007). These findings suggest that differential activation of Rap1 and Rap2 is important for dendritic development.

In *C. elegans* cholinergic motor neurons, *rap-2* prevents the overlap of adjacent presynaptic regions but does not alter overall synapse number (Chen et al., 2018). Here, we show that *rap-1* promotes synaptic vesicle accumulation and reduces the density of cholinergic synapses, which are not altered by *rap-2* (Fig. 2D,E),

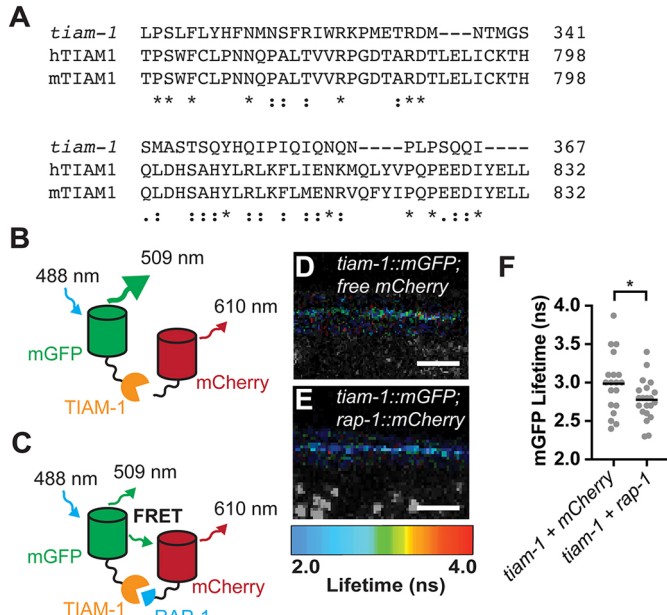

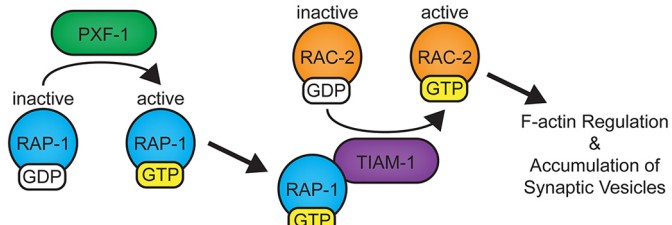

**Fig. 8. RAP-1 interacts with TIAM-1 in cholinergic motor neurons.**
(A) Protein sequence alignments of Ras-binding domains from human TIAM1 (hTIAM1) and mouse TIAM1 (mTIAM1) with *C. elegans* TIAM-1. Sequences were aligned using the Clustal Omega program (Sievers and Higgins, 2018; Sievers et al., 2011) through UniProt (UniProt Consortium, 2024). Degrees of amino acid conservation are indicated as an asterisk for fully conserved residues, a colon for conserved between residues with strongly similar properties, or a period for conserved between residues with weakly similar properties. (B,C) Schematic of interactions between TIAM-1::mGFP and free mCherry (B) and TIAM-1::mGFP and RAP-1::mCherry (C) expressed in cholinergic motor neurons under the *unc-17b* promoter used for FRET-FLIM experiments. Without RAP-1, there should be very little FRET between mGFP and mCherry (B), but if there is an interaction between RAP-1 and TIAM-1, FRET should reduce the lifetime of mGFP fluorescence at 509 nm (C). (D,E) FastFLIM images of mGFP in the dorsal cord of animals expressing each combination of plasmids in cholinergic neurons. The lifetime for mGFP has been pseudo-colored to represent fluorescence lifetime values in nanoseconds. Scale bars: 5 μm. (F) Mean lifetime fluorescence of TIAM-1::mGFP *in vivo* with control versus experimental plasmids. Data was pooled from multiple transgenes: TIAM-1::mGFP+mCherry (*bluEx148*, *bluEx149* and *bluEx150*) and TIAM-1::mGFP+RAP-1::mCherry (*bluEx152*, *bluEx153* and *bluEx154*). Gray circles represent individual animals, black bars indicate means. *n*=19-21. **P*<0.05 (unpaired two-tailed *t*-test).

**Fig. 9. Hypothetical model for the molecular mechanisms through which PXF-1 modulates synaptic vesicle accumulation.** During development, PXF-1 activates RAP-1 by stimulating the exchange of GDP for GTP. We propose that GTP-bound RAP-1 binds to TIAM-1. RAP-1 binding to TIAM-1 increases the local levels of RAC-2 activity by TIAM-1 stimulating the exchange of GDP for GTP. We hypothesize that GTP-bound RAC-2 interacts with the WAVE/SCAR complex required for actin filament formation. The presynaptic actin network then promotes the accumulation of synaptic vesicles at the nerve terminal.

further indicating a divergence of function at the synapse. The divergent effects of these Rap paralogs in some cases may be attributed to different signaling pathways. At postsynaptic sites, for example, Rap1 acts through AF-6 to influence dendritic spine neck length, but Rap2 opposes ERK signaling to reduce spine length (Ryu et al., 2008; Xie et al., 2005). At presynaptic terminals, *C. elegans* RAP-2 signals through TNIK to modulate the actin cytoskeleton (Chen et al., 2018), but RAP-1 appears to act through TIAM-1 and RAC-2 to promote stability of actin filaments (Figs 4,6,7). Identifying the mechanisms that differentially modulate Rap paralogs in presynaptic and postsynaptic compartments should reveal how these distinct signaling pathways coordinate synapse development and function.

### Regulation of Rac activity
Activation of RAP-1 was required to rescue *pxf-1* mutant deficits in synapse development; however, simply overexpressing RAC-2 was sufficient to restore synapse development in *pxf-1* mutants (Fig. 6).

This suggests that exogenous RAC-2 is activated by cholinergic neurons to a level sufficient to restore synapse assembly in *pxf-1* mutants. Mutations in *pxf-1* and *tiam-1* produce similar decreases in RAC-2 activity (Fig. 5). One possible reason that exogenous wild-type RAC-2 increases mCherry::RAB-3 intensity in *pxf-1* mutants could be that high levels of RAC-2 can be activated by another Rac GEF, like Vav, to compensate for the reduction in TIAM-1 signaling. Alternatively, a previous study has shown that inhibition of Rac GAP function in mammalian neurons increases Rac activity to a similar level as Tiam1 overexpression (Um et al., 2014). Therefore, another possible explanation for how overexpression of RAC-2 restores synaptic development in *pxf-1* mutants is that excess RAC-2 may act as a sponge for a Rac GAP, preventing it from terminating RAC-2 signaling, and allowing for sufficient levels of GTP-bound RAC-2 to promote synapse development. Additional studies will be required to determine how RAP-1 regulates TIAM-1 function and how the activity of Rac GEFs and GAPs may modulate RAC-2 activity downstream of PXF-1 signaling.

### Regulation of Rap GEF activity
GAPs and GEFs coordinate where and when G proteins are active. While protein-protein interactions oftentimes determine where GAPs and GEFs influence GTPase activity, it is not always clear if and how the activity of GAPs or GEFs is regulated. Some Ras and Rap GEFs are activated by second messengers, like calcium, diacylglycerol or cAMP (de Rooij et al., 2000, 1998; Ebinu et al., 1998; Kawasaki et al., 1998). Exchange-protein activated by cAMP (Epac) and Ras-associating (RA)-GEF/PDZ-GEF proteins constitute two very similar classes of Rap GEFs. Both contain cyclic nucleotide-binding domains, yet only Epacs are known to be activated by cAMP binding (de Rooij et al., 2000, 1998; Kawasaki et al., 1998). RA-GEF/PDZ-GEFs, like *pxf-1*, RAPGEF2 or RAPGEF6, do not appear to be modulated by cAMP or cGMP (Liao et al., 1999). Additional studies suggest that binding of Ras or Rap to these RA-GEF/PDZ-GEFs acts as a positive feedback loop to amplify the activity of these G proteins (Hu et al., 1999; Jin et al., 2001; Rebhun et al., 2000). Moreover, post-translational modification of Rap GEFs has been shown to modulate Rap GEF activity during neuronal migration (Ye et al., 2014); however, it is not known whether similar events occur during synapse development. Further studies will be required to determine whether PXF-1 and its homologs are modulated by post-translational modifications, the binding of cyclic nucleotide monophosphates, or additional G proteins during synapse development.

## MATERIALS AND METHODS

### DNA cloning and plasmid construction

To express RAP-1 in cholinergic neurons, *rap-1(WT)* cDNA was cloned by PCR with Q5 polymerase (New England Biolabs; NEB) using oSC188 and oSC189. We cloned *rap-1(G12V)* cDNA from pREA6 [*Prgef-1::rap-1 (G12V) cDNA*] (Addgene plasmid #245833) using oSC188 and oSC189. Using the Gibson assembly method, we inserted the *rap-1(WT)* PCR product into pBD23 [*pCR8-mGFP*] (Addgene plasmid #245830) digested with KpnI (NEB) to create pBD25 [*pCR8 rap-1(WT)::mGFP*] (Addgene plasmid #245765) and the *rap-1(G12V)* PCR product into pBD28 [*pCR8 tiam-1:: mGFP*] (Addgene plasmid #245766) digested with KpnI to create pSJC316 [*pCR8 rap-1(G12V)::mGFP*] (Addgene plasmid #245772). Using LR clonase II (Invitrogen), we inserted *rap-1(WT)::mGFP* or *rap-1(G12V):: mGFP* into pCZGY1091 [*Punc-17b-dest*] to create pBD31 [*Punc-17b::rap-1(WT)::mGFP*] (Addgene plasmid #245768) and pSJC319 [*Punc-17b::rap-1(G12V)::mGFP*] (Addgene plasmid #245775).

To express RAC-2 in cholinergic neurons, we cloned *rac-2(WT)* or *rac-2(G12V)* cDNA from pREA2 [*Prgef-1::rac-2(WT) cDNA*] (Addgene plasmid #245831) or pREA3 [*Prgef-1::rac-2(G12V) cDNA*] (Addgene plasmid #245832), respectively, by PCR with Q5 polymerase using oSC304 and oSC305. The *rac-2(WT)* or *rac-2(G12V)* PCR products were inserted into pBD28 [*pCR8 tiam-1::mGFP*] digested with KpnI to create pSJC317 [*pCR8 rac-2(WT)::mGFP*] (Addgene plasmid #245773) and pSJC318 [*pCR8 rac-2(G12V)::mGFP*] (Addgene plasmid #245774). Using LR clonase II, we inserted the *rac-2(WT)::mGFP* or *rac-2(G12V)::mGFP* into pCZGY1091 [*Punc-17b-dest*] to create pSJC320 [*Punc-17b::rac-2(WT):: mGFP*] (Addgene plasmid #245776) and pSJC321 [*Punc-17b::rac-2(G12V)::mGFP*] (Addgene plasmid #245777).

For cholinergic expression of the RAC-2 activity biosensor, we created pREA12 [*pCR8 mCherry::RBD::T2A::rac-2::mGFP*] (Addgene plasmid #245770) using Gibson assembly between pSJC213, a modified pCR8 Gateway entry vector, digested with NdeI (NEB) and two gBlocks (Integrated DNA Technologies) encoding *mCherry::RBD::rac-2 N-terminus*, which includes the RBD from *pak-1* and a T2A ribosomal skip sequence, and *rac-2 C-terminus::mGFP*. pREA16 [*Punc-17b::mCherry::RBD::T2A::rac-2:: mGFP*] (Addgene plasmid #245771) was generated using an LR reaction between pREA12 and pCZGY1091.

To measure protein-protein interactions between RAP-1::mCherry and TIAM-1::mGFP, we created pBD28 [*pCR8 tiam-1::mGFP*] using Gibson assembly between two gBlocks encoding the N-terminus and C-terminus of *tiam-1* cDNA and pBD23 digested with KpnI. pBD24 [*pCR8 rap-1::mCherry*] (Addgene plasmid #245764) was created by Gibson assembly between pBD22 [*pCR8 mCherry*] (Addgene plasmid #245763) digested with KpnI and *rap-1* cDNA amplified by PCR as described above. We used LR clonase II to insert *tiam-1::mGFP* or *rap-1::mCherry* into pCZGY1091 to create pBD34 [*Punc-17b::tiam-1::mGFP*] (Addgene plasmid #245769) and pBD30 [*Punc-17b:: rap-1::mCherry*] (Addgene plasmid #245767). All entry vector sequences were confirmed by Sanger sequencing (Eurofins). Plasmid design and sequencing results were analyzed using ApE (Davis and Jorgensen, 2022).

### *C. elegans* strains, maintenance, and transgenesis

All studies were performed using *C. elegans* hermaphrodites maintained on standard nematode growth medium plates (NGM) at 20°C and seeded with OP50 (*Caenorhabditis* Genetics Center; WormBaseID: OP50) as previously described (Brenner, 1974). All transgenic strains were generated in the lab using the previously described microinjection technique (Mello et al., 1991) using *Pmyo-2::mCherry* (2 ng/μl) or *Punc-122::RFP* (60 ng/μl) co-injection markers. Strain details are listed in Table S1. Strains generated in this study can be obtained by contacting the corresponding author. Genetic mutants were identified by genotyping with the following primer sets: oSC48 and oSC49 for *pxf-1(gk955083)*; oSC236 and oSC237 for *rap-1(pk2082)*; oRL21, oRL22 and oRL23 for *rap-2(gk11)*; oRL4, oRL5 and oRL6 for *rac-2(ok326)*; oBM65, oBM66 and oBM63 for *tiam-1(tm1556)*; oBM62, oBM63 and oBM64 for *tiam-1(ok772)*. Primer sequences are listed in Table S2.

### Aldicarb behavioral assay

Day one adults were placed on NGM plates containing 1 mM aldicarb (Sigma-Aldrich) and scored for paralysis at 15-min intervals during 2 h of observation. Plates were prepared and poured in batches and kept at 4°C before use. Paralysis was defined as no response after three touches. Plates were placed on the bench 30 min before the experiments, and all assays were performed at room temperature (20-21°C). Individuals performing the assays were unaware of the genotypes.

### Fluorescence microscopy

To visualize changes to synaptic vesicle pools, mutant animals were crossed with Punc-129::mCherry::RAB-3(*tauIs46*) synaptic vesicle marker. To visualize filamentous actin, mutant animals were crossed with mGFP fused to the calponin homology domain of Ut (*bluEx30*) (Lamb et al., 2022). Animals were immobilized in M9 buffer on 10% agarose pads. Images were taken using a DS-Qi2 camera at 60× magnification with a 1.20 NA water immersion lens using Ti-2E widefield fluorescent microscope (Nikon) along the posterior dorsal cord in L4 animals and analyzed using the FIJI distribution of NIH ImageJ (Schindelin et al., 2012) as previously described (Lamb et al., 2022).

### Whole mount immunohistochemistry

To visualize endogenous expression of UNC-17, adult animals were fixed as previously described (Cherra and Jin, 2016; Lamb et al., 2022). Samples were blocked with 5% bovine serum albumin (BSA) for 1 h and incubated overnight with an UNC-17 antibody (1:500, Mab1403, Developmental Studies Hybridoma Bank, AB_2315531). Samples were washed with 2% BSA in phosphate buffered saline with Triton X-100 (PBST) and incubated for 1 h with Alexa Fluor 594 donkey anti-mouse secondary antibody (Thermo Fisher Scientific, AB_141633, 1:2000). After washing, samples were mounted using Vectashield mounting medium (VWR), covered with a coverslip, and imaged using the Ti2-E microscope (Nikon) at 60× magnification with a 1.20NA water immersion lens (Nikon).

### Fluorescence lifetime imaging microscopy

To visualize *in vivo* changes to RAC-2 activity or interactions between RAP-1 and TIAM-1, L4 animals containing FRET-FLIM biosensors expressed in cholinergic neurons were immobilized in M9 buffer on 10% agarose pads. FRET-FLIM biosensors were constructed using the mGFP-mCherry donor acceptor pairing under the *unc-17b* promoter. Cholinergic synapses were located along the dorsal cord using the FITC laser on a Nikon A1 Confocal microscope. Images were acquired for 1 min at 20°C using a 60×1.20NA water immersion lens (Nikon) using a PMA 40 Hybrid Photomultiplier Detector (Picoquant) and a SuperK Extreme white light laser (NKT Phototonics) equipped with a 488 nm filter. The laser was pulsing at 38.9 MHz. Output signal was measured using a Picoharp 300 Time-Correlated Single Photon Counting system at 8.0 ps resolution (Picoquant). All frames were binned together. Mean fluorescence lifetimes of mGFP molecules were calculated using PicoQuant SymPho Time 64 software.

### Data quantification and statistical analysis

All statistical analyses were performed using Prism 9 (GraphPad). For comparisons between two groups, we used a *t*-test when distributed normally and a Mann–Whitney test for non-parametric data. For comparisons between multiple groups with normally distributed data, a one-way ANOVA was performed followed by Sidak's multiple comparisons test. For data that was not normally distributed, a Kruskal–Wallis test was used followed by Dunn's multiple comparisons test. We used a mixed-effects analysis followed by Dunnett's multiple comparisons for drug-induced paralysis time courses. A corrected *P*-value less than 0.05 was considered significant. Additional statistical information for individual figures may be found within the legends.

### Acknowledgements

We thank Avanti Sawardekar and Bithika Dhar for the creation of reagents and initial validation of *rap-1* and *rac-2* cDNA expression. FRET-FLIM image acquisition and data analysis were performed using resources at the Light Microscopy Core at the University of Kentucky. Some mutant strains were provided by the *Caenorhabditis* Genetics Center, which is supported by the NIH (OD010440).

### Competing interests

The authors declare no competing or financial interests.

## Author contributions
Conceptualization: R.L., S.J.C.; Formal analysis: R.L., M.S., J.W., M.W., S.J.C.; Funding acquisition: R.L., S.J.C.; Investigation: R.L., M.S., J.W., M.W., S.J.C.; Project administration: S.J.C.; Writing – original draft: R.L., S.J.C.; Writing – review & editing: R.L., M.S., J.W., M.W., S.J.C.

## Funding
This research was supported in part by grants from the National Institute of Neurological Disorders and Stroke (NS097638 and NS129668-01A1 to S.J.C., NS129159-01A1 to R.L.). Open Access funding provided by University of Kentucky. Deposited in PMC for immediate release.

## Data and resource availability
All relevant data and resources can be found within the article and its supplementary information. ImageJ script is available on GitHub (github.com/samcherra/Synapse_OneColorLineScan). Plasmids generated in this study have been deposited into the Addgene repository (see Materials and Methods).

## The people behind the papers
This article has an associated 'The people behind the papers' interview with some of the authors.

## Peer review history
The peer review history is available online at https://journals.biologists.com/dev/lookup/doi/10.1242/dev.204678.reviewer-comments.pdf

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
