## [Peer Review File · Development (Cambridge, England)]

A PDZ-RapGEF promotes synaptic development in *Caenorhabditis elegans* through a Rap/Rac signaling pathway

Reagan Lamb, Michael Scales, Julie Watkins, Martin Werner and Salvatore J. Cherra, III
DOI: 10.1242/dev.204678

Editor: Steve Wilson

Review timeline

Original submission:	24 January 2025
Editorial decision:	31 March 2025
First revision received:	27 June 2025
Accepted:	25 July 2025

Original submission

First decision letter

MS ID#: dev.204678

MS TITLE: A PDZ-RapGEF promotes synaptic development in *C. elegans* through a Rap/Rac signaling pathway

AUTHORS: Reagan Lamb, Michael Scales, Julie Watkins, Martin Werner and Salvatore James Cherra

Dear Salvatore,

Apologies for the delay in obtaining reviews on your manuscript. However, I have now received two referees' reports on the above manuscript, and have reached a decision. The referees' comments are appended below, or you can access them online: please go to:

As you will see, the referees express considerable interest in your work, but have some suggestions for improving the manuscript. If you are able to revise the manuscript along the lines suggested, I will be happy receive a revised version of the manuscript. Please also note that Development will normally permit only one round of major revision. If it would be helpful, you are welcome to contact us to discuss your revision in greater detail. Please send us a point-by-point response indicating your plans for addressing the referees' comments, and we will look over this and provide further guidance.

Please attend to all of the reviewers' comments and ensure that you clearly highlight all changes made in the revised manuscript. Please avoid using 'Tracked changes' in Word files as these are lost in PDF conversion. I should be grateful if you would also provide a point-by-point response detailing how you have dealt with the points raised by the reviewers in the 'Response to Reviewers' box. If you do not agree with any of their criticisms or suggestions please explain clearly why this is so.

Reviewer 1

Advance summary and potential significance to field

This work defines a cascade of conserved GTPases and their GEF modulators that promote synaptic vesicle accumulation at cholinergic synapses. Findings are robust and thus are expected to spur investigations in more complex organisms to test the pathway defined here.

Comments for the author

This work uses genetic and imaging analysis in *C. elegans* to identify downstream effectors of a PDZ-GEF protein for assembly of synaptic vesicles at cholinergic synapses. In previous work, this laboratory determined that PXF-1, a PDZ-GEF, is required for normal accumulation of synaptic vesicles in cholinergic neurons. Because GEFs are predicted to activate members of the RAS family of small GTPases, the authors set out to identify specific GTPases that might function downstream of PXF-1 for synaptic vesicle (SV) localization. Initial experiments confirm that *pxf-1* mutants retard the accumulation of the SV marker, RAB-3, in developing cholinergic motor neurons. Transgenic expression of a constitutively active form of RAP-1 GTPase rescues the *pxf-1* mutant, consistent with a downstream role for RAP-1. Additional genetic experiments also place RAC-2/GTPase in the PXF-1 pathway. A downstream role for RAC-2 is suggested by the results of an *in vivo* FRET assay showing that RAC-2 binding to a canonical RAS binding domains depends on *pxf-1*. A downstream role for RAC-2 is also consistent with the finding that a gain of function RAC-2 construct rescues both the *pxf-1* and *rap-2* SV defects. Finally, an *in vivo* FRET experiment suggests that RAP-1 physically interacts with TIAM-1 and thereby potentially functions as a RAP-1 effector for activating RAC-2 (see below). Genetic results are presented in a logical order, the writing is clear (see below for suggested minor revisions) and the overall finding of a GTPase cascade for synaptic assembly is well supported. The use of *in vivo* FRET assays to detect physical interactions is a strength of this work. The proposed RAP-1 dependent activation of TIAM-GEF activity merits clarification, potentially with additional biochemical experiments to confirm this model (see below).
Major revisions.

GTPases can function in signaling cascades that activate other downstream GTPases. This paper uses a FRET assay to demonstrate that RAP-1 binds TIAM-1-GEF *in vivo* and proposes that this interaction could in turn activate TIAM-1 to function as a GEF for RAC-2. Is this a novel finding, that a RAS protein can function as a direct activator of TIAM-GEF activity? Can you use a biochemical assay to confirm that RAP-1 binding enhances TIAM GEF activity? Does proposed RAS binding domain in TIAM resemble other RBDs in alpha fold models?

Minor Revisions.

The paper needs a graphical abstract to help readers keep track of the biochemical identity and position of players (PXF-1, RAP-1, RAC-2, TIAM-1) in the proposed pathway. Each figure could also include a schematic of each segment of the pathway as it is deduced from successive experiments. I had to draw out the pathway while reading the manuscript to keep track of key findings.

Fig 1. What is the promoter for the RAB-3-mCherry marker in Figure 1?

Ln 101-102. What is significance of the increase in the number of RAB-3 labeled puncta in a *rap-1* mutant? Smaller more numerous puncta?

Ln118-130 Isn't WVE-2 a conserved component of the WAVE/SCAR complex? This is confusing. Please explain the interpretation of the aldicarb experiment, e.g., aldicarb antagonizes acetylcholinesterase leading to elevated levels of ACh and hyper-muscle contraction...Please define UNC-17 (i.e., Vesicular Acetyl Choline Transporter, VACht)

Figure 6A Data for wild type control and *pxf-1* mutant are indistinguishable; the key shows a solid dot for control and a dot with open center for *pxf-1* but all dots in the figure are solid. Better to use different colors for control vs *pxf-1* in any case. The use of color to distinguish genotypes would also help readers interpret other figures like this one.

Lin253-256. "Since the small GTPases and regulatory proteins investigated here do not show a synaptogenesis phenotype..." This statement is confusing since multiple figures show synaptogenesis defects for GTPases and GEFs?

Lns 283-288. I don't follow this argument. Please simplify and explain

Demarco RS, Struckhoff EC, Lundquist EA. 2012. The Rac GTP exchange factor TIAM-1 acts with CDC-42 and the guidance receptor UNC-40/DCC in neuronal protrusion and axon guidance. *PLoS Genet.* 8(4):e1002665.

Tang LT, D'az-Balzac CA, Rahman M, Ramirez-Suarez NJ, Salzberg Y, Lazaro-Pena MI, B'low HE. 2019. TIAM-1/GEF can shape somatosensory dendrites independently of its GEF activity by regulating F-actin localization. *eLife.* 8:507.

Zhu Y, Tesone Z, Tan M, Hardin J. 2024. TIAM-1 regulates polarized protrusions during dorsal intercalation in the *Caenorhabditis elegans* embryo through both its GEF and N-terminal domains. *J Cell Sci.* 137(5):jcs261509. doi:10.1242/jcs.261509.

Reviewer 2

Advance summary and potential significance to field

This work shows that in *C. elegans*, the RAP-1 GTPase acts downstream of PXF-1 to promote development of pre-synaptic terminals at the neuromuscular junction. Activation of RAP-1 by PXF-1 leads in turn to activation of the small GTPase RAC-2, which modulates the actin cytoskeleton at pre-synaptic regions. Activated RAP-1 appears to stimulate RAC-2 by interacting with the RAC-2 GEF TIAM-1. These findings are supported by detailed phenotypic analyses of single and double mutants of these genes using an mCherry::RAB-3 reporter for synapses in the dorsal cord of *C. elegans*, and by FRET-FLIM studies of RAC-2 activity and TIAM-1 RAP-1 interactions.

Previous work in mammalian cells had provided evidence of a similar regulatory interaction between Rap and Rac GTPases mediated via Tiam1. This work shows that this cassette promotes accumulation of synaptic vesicles at presynaptic terminals.

Comments for the author

In general the manuscript is clearly written, and the experiments straightforward, well described and easy to understand. In some places the authors do not provide readers enough background information to fully understand their experiments. I indicate where this is the case below, and suggest they add one or two sentences to remedy these gaps. In a few places the authors make strong statements on the basis of comparing the phenotypes of single and double mutants. I suggest these assertions be softened.

Minor suggestions:

L102: would be clearer to say '...a small but significant increase in the number of RAB-3 -labeled puncta....'.

L120: The authors need to introduce WVE-1 better.

L131: The authors could add a sentence justifying why they chose to work on RAC-2 instead of CED-10, which appears to have a stronger phenotype.

L134: The authors need to better introduce the mGFP::ut::CH reagent they use to label filamentous actin.

L148-9: '...was not significantly different from either single mutants (Figure 4A-E), indicating that PXF-1 and RAC-2 act in the same pathway.' Indicating is a strong word to use here. It would be better to say 'consistent with' or 'suggesting that'.

L168: '...to determine if pxf-1 mutants affected the level of RAC-2 activity....'. I suggest rephrasing this to say: 'To determine if disrupting PXF-1 altered the level of RAC-2 activity, we compared mGFP::RAC-2 lifetime fluorescence in pxf-1 mutants and wild type animals.'

L212: There were no changes in synapse number in any single or double mutants (Figure 7F and 7L) indicating that.... I suggest replacing indicating that by 'consistent with' or 'suggesting that'.

Fig 5A. The authors could explicitly illustrate the FLIM assay.

Fig 6A. The figure seems to be incorrect in that only filled gray dots, representing wild type, are shown. Open dots representing pxf-1 mutants are absent.

Fig. 7E, F, K, L: The authors should add statistics for comparisons between wild type and the double mutant.

First revision

Author response to reviewers' comments

Thank you for the critiques and suggestions. We believe they have strengthened the message of the manuscript. We have indicated all changes in the manuscript as blue text. Our responses to each concern or suggestion is indicated below in the same blue colored text.

Comments from the Reviewers:

Reviewer 1: SUMMARY OF THE ADVANCE MADE IN THIS PAPER AND ITS POTENTIAL SIGNIFICANCE TO THE FIELD

This work defines a cascade of conserved GTPases and their GEF modulators that promote synaptic vesicle accumulation at cholinergic synapses. Findings are robust and thus are expected to spur investigations in more complex organisms to test the pathway defined here.

SUGGESTIONS TO AUTHORS

This work uses genetic and imaging analysis in *C. elegans* to identify downstream effectors of a PDZ-GEF protein for assembly of synaptic vesicles at cholinergic synapses. In previous work, this laboratory determined that PXF-1, a PDZ-GEF, is required for normal accumulation of synaptic vesicles in cholinergic neurons. Because GEFs are predicted to activate members of the RAS family of small GTPases, the authors set out to identify specific GTPases that might function downstream of PXF-1 for synaptic vesicle (SV) localization. Initial experiments confirm that *pxf-1* mutants retard the accumulation of the SV marker, RAB-3, in developing cholinergic motor neurons. Transgenic expression of a constitutively active form of RAP-1 GTPase rescues the *pxf-1* mutant, consistent with a downstream role for RAP-1. Additional genetic experiments also place RAC-2/GTPase in the PXF-1 pathway. A downstream role for RAC-2 is suggested by the results of an *in vivo* FRET assay showing that RAC-2 binding to a canonical RAS binding domains depends on *pxf-1*. A downstream role for RAC-2 is also consistent with the finding that a gain of function RAC-2 construct rescues both the *pxf-1* and *rap-2* SV defects. Finally, an *in vivo* FRET experiment suggests that RAP-1 physically interacts with TIAM-1 and thereby potentially functions as a RAP-1 effector for activating RAC-2 (see below). Genetic results are presented in a logical order, the writing is clear (see below for suggested minor revisions) and the overall finding of a GTPase cascade for synaptic assembly is well supported. The use of *in vivo* FRET assays to detect physical interactions is a strength of this work. The proposed RAP-1 dependent activation of TIAM-GEF activity merits clarification, potentially with additional biochemical experiments to confirm this model (see below). Major revisions.

GTPases can function in signaling cascades that activate other downstream GTPases. This paper uses a FRET assay to demonstrate that RAP-1 binds TIAM-1-GEF *in vivo* and proposes that this interaction could in turn activate TIAM-1 to function as a GEF for RAC-2. Is this a novel finding, that a RAS protein can function as a direct activator of TIAM-GEF activity? Can you use a biochemical assay to confirm that RAP-1 binding enhances TIAM GEF activity? Does proposed RAS binding domain in TIAM resemble other RBDs in alpha fold models?

Rap GTPases previously have been found to activate Rac GEFs. Previous studies have linked Rap GTPases to directly activating Vav and Tiam1 in non-neuronal cells. A few studies in neurons have shown that Rap1 can interact with Vav. These references to the current status of the field are included in the discussion.

Previous studies showed that Rap binding recruits Vav (Arthur et al. 2004) or Tiam1 (Birukova et al. 2013) to the plasma membrane where they stimulate Rac signaling. To test whether RAP-1

binding activates TIAM-1 GEF activity towards RAC-2, we have expressed each C. elegans protein in bacteria. We have successfully purified RAP-1 and RAC-2 C. elegans proteins from bacteria; however, we were unable to purify TIAM-1, even though the bacteria displayed robust expression upon induction. There appears to be an issue with solubility or folding that is hindering the binding of GST-TIAM-1 to the purification column. Previous studies have trimmed portions off TIAM-1 homologs for purification, but those approaches remove domains that alter TIAM-1 activity. At this point, we cannot perform an in vitro biochemical assay to measure the direct activity of TIAM-1 in the presence or absence of RAP-1 binding without extensive optimization of TIAM-1 expression and purification protocols, which, if possible, are beyond our current capabilities.

We analyzed the putative RBD in C. elegans TIAM-1 using pairwise structure alignment (www.rcsb.org; H.M. Berman, J. Westbrook, Z. Feng, G. Gilliland, T.N. Bhat, H. Weissig, I.N. Shindyalov, P.E. Bourne, The Protein Data Bank (2000) Nucleic Acids Research 28: 235-242 <https://doi.org/10.1093/nar/28.1.235>). We compared AlphaFold structures of TIAM-1 to RBDs from human Tiam1, Tiam2, Raf, and Regulator of G-protein signaling 12 and the Ras-associating (RA) domains from RapGEF2, RalGEF, and Afadin. Among TIAM-1 homologs, the best alignments with the lowest root mean squared deviations (RMSD) below 3 and TM-Scores above 0.7 were derived from peptides containing the human PDZ domain. This suggests that C. elegans TIAM-1 most likely contains a PDZ domain, despite poor sequence alignments for this region. In human Tiam1, the PDZ domain is preceded by the RBD. Unfortunately, the region preceding the predicted PDZ of C. elegans TIAM-1 has low to very low confidence values for structural prediction in AlphaFold. Therefore, our analyses were limited by available structural predictions. Future experiments beyond the scope of this study will need to analyze the function of this region in TIAM-1 using mutational analyses or chimeric proteins, where human and C. elegans RBD domains are swapped, to fully test whether the interaction between RAP-1 and TIAM-1 in C. elegans proceeds in a similar manner as the mammalian proteins.

Minor Revisions.

The paper needs a graphical abstract to help readers keep track of the biochemical identity and position of players (PXF-1, RAP-1, RAC-2, TIAM-1) in the proposed pathway. Each figure could also include a schematic of each segment of the pathway as it is deduced from successive experiments. I had to draw out the pathway while reading the manuscript to keep track of key findings.

We have added a hypothetical model that summarizes our interpretations of the data presented in the revised manuscript. The model is displayed as Figure 9.

Fig 1. What is the promoter for the RAB-3-mCherry marker in Figure 1?

The transgene expressing mCherry::RAB-3 is from the unc-129 promoter. This information is included in the Table 1, but we have now included the information in the text and figure legend describing Figure 1.

Ln 101-102. What is significance of the increase in the number of RAB-3 labeled puncta in a rap-1 mutant? Smaller more numerous puncta?

The rap-1 mutants appear to have dimmer more numerous puncta. We currently are not sure what this could mean biologically. For example, it could represent increased synapse formation, but reduced maturation/development. While pxf-1 mutants cause a reduction in maturation/development, another RapGEF may be responsible for the increased synaptogenesis. As stated in the discussion, we are not sure about the underlying causes for this rap-1 mutant phenotype.

Ln118-130 Isn't WVE-2 a conserved component of the WAVE/SCAR complex? This is confusing. Please explain the interpretation of the aldicarb experiment, e.g., aldicarb antagonizes acetylcholinesterase leading to elevated levels of ACh and hyper-muscle contraction...Please define UNC-17 (i.e., Vesicular Acetyl Choline Transporter, VACht)

WVE-1 is a component of the conserved WAVE/SCAR complex. We previously showed that presynaptic levels of F-actin are reduced in pxf-1 mutants (Lamb et al. 2022). We found that over expression of WVE-1 in neurons was sufficient to restore mCherry::RAB-3 levels in pxf-1 mutants. Based on these results, we hypothesized that a Rac GTPase might act downstream of PXF-1 and RAP-1 to promote synapse development. We have clarified these points in the revised text.

We have revised the text to explain the effects of aldicarb as an acetylcholinesterase inhibitor. Exposure to aldicarb leads to a buildup of acetylcholine at the neuromuscular junction that results in paralysis. As compared to wild type animals, rac-2 and ced-10 mutants display a reduced rate of paralysis. These results are consistent with rac-2 and ced-10 being required for proper synaptic transmission at the neuromuscular junction, like pxf-1.

We have revised the text to define UNC-17 as the worm homolog of the vesicular acetylcholine transporter.

Figure 6A Data for wild type control and pxf-1 mutant are indistinguishable; the key shows a solid dot for control and a dot with open center for pxf-1 but all dots in the figure are solid. Better to use different colors for control vs pxf-1 in any case. The use of color to distinguish genotypes would also help readers interpret other figures like this one.

Indeed, the symbols were not properly formatted when we switched the graph to a grayscale representation. The revised graph now includes red circles for the pxf-1 mutant symbols. We have also revised Figure 6B so rac-2 mutants are represented as blue circles.

Lin253-256. "Since the small GTPases and regulatory proteins investigated here do not show a synaptogenesis phenotype..." This statement is confusing since multiple figures show synaptogenesis defects for GTPases and GEFs?

We have clarified the text to identify RAC-2 and TIAM-1, which were the GTPase and regulatory proteins that act downstream of RAP-1 but do not cause increases in synapse number (synaptogenesis effect). Neither rac-2 nor tiam-1 mutants display a change in synapse number (synaptogenesis), but both show a reduction in synaptic vesicles (synapse assembly/development/maturation).

Lns 283-288. I don't follow this argument. Please simplify and explain

We have revised this section as follows: "This suggests that exogenous RAC-2 is activated by cholinergic neurons to a level sufficient to restore synapse assembly in pxf-1 mutants. Mutations in pxf-1 and tiam-1 produce similar decreases in RAC-2 activity (Figure 5). One possible reason that exogenous wild type RAC-2 increases mCherry::RAB-3 intensity in pxf-1 mutants could be that high levels of RAC-2 can be activated by another Rac GEF, like Vav, to compensate for the reduction in TIAM-1 signaling. Alternatively, a previous study showed that inhibition of Rac GAP function in mammalian neurons increases Rac activity to a similar level as Tiam1 overexpression (Um et al., 2014). Therefore, another possible explanation for how overexpression of RAC-2 restores synaptic development in pxf-1 mutants is that excess RAC-2 may act as a sponge for a Rac GAP, preventing it from terminating RAC-2 signaling, and allowing for sufficient levels of GTP-bound RAC-2 to promote synapse development. Additional studies will be required to determine how RAP-1 regulates TIAM-1 function and how the activity of Rac GEFs and GAPs may modulate RAC-2 activity downstream of PXF-1 signaling.

Demarco RS, Struckhoff EC, Lundquist EA. 2012. The Rac GTP exchange factor TIAM-1 acts with CDC-42 and the guidance receptor UNC-40/DCC in neuronal protrusion and axon guidance. *PLoS Genet.* 8(4):e1002665.

Tang LT, Díaz-Balzac CA, Rahman M, Ramirez-Suarez NJ, Salzberg Y, Lazaro-Pena MI, Bülow HE. 2019. TIAM-1/GEF can shape somatosensory dendrites independently of its GEF activity by regulating F-actin localization. *eLife.* 8:507.

Zhu Y, Tesone Z, Tan M, Hardin J. 2024. TIAM-1 regulates polarized protrusions during dorsal intercalation in the *Caenorhabditis elegans* embryo through both its GEF and N-terminal domains. *J Cell Sci.* 137(5):jcs261509. doi:10.1242/jcs.261509.

Reviewer 2: SUMMARY OF THE ADVANCE MADE IN THIS PAPER AND ITS POTENTIAL SIGNIFICANCE TO THE FIELD

This work shows that in *C. elegans*, the RAP-1 GTPase acts downstream of PXF-1 to promote development of pre-synaptic terminals at the neuromuscular junction. Activation of RAP-1 by PXF-1 leads in turn to activation of the small GTPase RAC-2, which modulates the actin cytoskeleton at pre-synaptic regions. Activated RAP-1 appears to stimulate RAC-2 by interacting with the RAC-2 GEF TIAM-1. These findings are supported by detailed phenotypic analyses of single and double mutants of these genes using an mCherry::RAB-3 reporter for synapses in the dorsal cord of *C. elegans*, and by FRET-FLIM studies of RAC-2 activity and TIAM-1 RAP-1 interactions.

Previous work in mammalian cells had provided evidence of a similar regulatory interaction between Rap and Rac GTPases mediated via Tiam1. This work shows that this cassette promotes accumulation of synaptic vesicles at presynaptic terminals.

SUGGESTIONS TO AUTHORS

In general the manuscript is clearly written, and the experiments straightforward, well described and easy to understand. In some places the authors do not provide readers enough background information to fully understand their experiments. I indicate where this is the case below, and suggest they add one or two sentences to remedy these gaps. In a few places the authors make strong statements on the basis of comparing the phenotypes of single and double mutants. I suggest these assertions be softened.

Minor suggestions:

L102: would be clearer to say '...a small but significant increase in the number of RAB-3 -labeled puncta....'.

We have revised the statement as suggested.

L120: The authors need to introduce WVE-1 better.

We have provided a more detailed explanation of WVE-1, its function at the synapse, and how we used our previous findings with WVE-1 overexpression to link pxf-1 to Rac GTPase function.

L131: The authors could add a sentence justifying why they chose to work on RAC-2 instead of CED-10, which appears to have a stronger phenotype.

We chose to work on RAC-2 since less is known about its function as compared to CED-10. We will note this in the text. The phenotypes are not different statistically between the ced-10 and rac-2 alleles.

L134: The authors need to better introduce the mGFP::ut::CH reagent they use to label filamentous actin.

We now provide additional details about utrophin, its interaction with filamentous actin, the original works that developed initial reagents in other model systems, and its use to measure filamentous actin at presynaptic terminals.

L148-9: '...was not significantly different from either single mutants (Figure 4A-E), indicating that PXF-1 and RAC-2 act in the same pathway.' Indicating is a strong word to use here. It would be better to say 'consistent with' or 'suggesting that'.

We have revised the text to read "consistent with".

L168: '...to determine if pxf-1 mutants affected the level of RAC-2 activity....'. I suggest rephrasing this to say: 'To determine if disrupting PXF-1 altered the level of RAC-2 activity, we compared mGFP::RAC-2 lifetime fluorescence in pxf-1 mutants and wild type animals.'

We have revised the structure of the sentence as suggested.

L212: 'There were no changes in synapse number in any single or double mutants (Figure 7F and 7L) indicating that...'. I suggest replacing indicating that by 'consistent with' or 'suggesting that'.

We have revised to “consistent with”.

Fig 5A. The authors could explicitly illustrate the FLIM assay.

We have added a panel illustrating the protein constructs and how they are expected to interact based on wild type and mutant backgrounds, where lower levels of RAC-2 activity reduce FRET between mGFP and mCherry fusion proteins that causes an increase in mGFP lifetime.

Fig 6A. The figure seems to be incorrect in that only filled gray dots, representing wild type, are shown. Open dots representing pxf-1 mutants are absent.

We have corrected the symbols.

Fig. 7E, F, K, L: The authors should add statistics for comparisons between wild type and the double mutant.

These comparisons have been added to the revised manuscript.

Second decision letter

MS ID#: dev.204678R1

MS TITLE: A PDZ-RapGEF promotes synaptic development in *C. elegans* through a Rap/Rac signaling pathway

AUTHORS: Reagan Lamb, Michael Scales, Julie Watkins, Martin Werner and Salvatore James Cherra

Dear Salvatore,

I am happy to tell you that the reviewers are happy with your revisions and your manuscript has been accepted for publication in Development, pending our standard publication integrity checks.

Reviewer 1

Advance summary and potential significance to field

see previous review

Comments for the author

the authors have fully addressed my concerns

Reviewer 2

Advance summary and potential significance to field

The authors have addressed my concerns.